# Evaluation of Direct Detection Protocols for Poliovirus from Stool Samples of Acute Flaccid Paralysis Patients

**DOI:** 10.3390/v15102113

**Published:** 2023-10-18

**Authors:** Minami Kikuchi Ueno, Kouichi Kitamura, Yorihiro Nishimura, Minetaro Arita

**Affiliations:** Department of Virology II, National Institute of Infectious Diseases, 4-7-1 Gakuen, Musashimurayama, Tokyo 208-0011, Japan; mi-kikuchi@niid.go.jp (M.K.U.); kkita@niid.go.jp (K.K.); ynishi@niid.go.jp (Y.N.)

**Keywords:** poliovirus, non-polio enterovirus, Global Polio Laboratory Network, direct detection, nanopore sequencing

## Abstract

Polio surveillance in the Global Polio Eradication Initiative has been conducted with virus isolation from stool samples of acute flaccid paralysis (AFP) cases. Under the current biorisk management/regulations, challenges arise in the timelines of the report, sensitivity of the test and containment of poliovirus (PV) isolates. In the present study, we evaluated protocols of previously reported direct detection (DD) methods targeting the VP1 or VP4–VP2 regions of the PV genome in terms of sensitivity and sequencability. An optimized protocol targeting the entire-capsid region for the VP1 sequencing showed a high sensitivity (limit of detection = 82 copies of PV genome) with a simpler and faster reaction than reported ones (i.e., with the addition of all the primers at the start of the reaction, the RT-PCR reaction finishes within 2.5 h). The DD methods targeting the VP1 region detected PV in 60 to 80% of PV-positive stool samples from AFP cases; however, minor populations of PV strains in the samples with virus mixtures were missed by the methods. Sequencability of the DD methods was primarily determined by the efficiency of the PCRs for both Sanger and nanopore sequencing. The DD method targeting the VP4–VP2 region showed higher sensitivity than that targeting the VP1 region (limit of detection = 25 copies of PV genome) and successfully detected PV from all the stool samples examined. These results suggest that DD methods are effective for the detection of PV and that further improvement of the sensitivity is essential to serve as an alternative to the current polio surveillance algorithm.

## 1. Introduction

The World Health Organization (WHO) leads the Global Polio Eradication Initiative (GPEI) and sets goals for stopping poliovirus (PV) transmission globally [1]. To achieve the goals, global surveillance for detecting PV from stool samples of acute flaccid paralysis (AFP) cases is essential to monitor the progress of PV eradication [2].

In the current surveillance algorithm in the Global Polio Laboratory Network (GPLN), virus isolation (VI) serves as the gold standard for PV detection [3]. After VI, intratypic differentiation (ITD) with real-time reverse transcription-PCR (rRT-PCR) and Sanger sequencing is conducted to identify the types of PV, i.e., wild-type PV (WPV), vaccine-derived PV (VDPV) and vaccine-like PV [3]. VI serves as a sensitive and affordable test and provides PV isolates for subsequent tests, including ITD. However, VI has drawbacks in the timeliness of reporting (10 days to confirm the negative results) and sensitivity [4]. Moreover, isolation of PV raises a concern against the current global containment plan for PV under the fourth edition of the WHO Global Action Plan to minimize poliovirus facility-associated risk after type-specific eradication of wild polioviruses and sequential cessation of oral polio vaccine use (GAP IV). Therefore, the WHO encourages the development of cell-culture-independent direct detection (DD) methods for PV from the stool samples.

To date, the DD methods targeting the entire-capsid region followed by VP1 sequencing [5,6] or the VP4–VP2 region [7,8,9,10] have been reported. While the analysis of the VP1 region is essential for the ITD, which could detect circulating VDPV (cVDPV) and trigger a vaccine campaign [11,12,13], the VP4–VP2 region is useful for sensitive detection of enteroviruses, including PV [7,9,10]. In this study, we evaluated protocols of the reported DD methods in terms of the sensitivity and sequencability by using PV-positive stool samples from AFP cases. The potential of the DD methods as an alternative to the current algorithm are discussed below.

## 2. Materials and Methods

**Virus and stool samples.** Viral RNA genomes of PV1(Sabin 1) (GenBank: AY184219), PV2(Sabin 2) (GenBank: AY184220) and PV3(Sabin 3) (GenBank: AY184221) strains, and EV-D68(US/KY/14-18953) (GenBank: KM851231) and EV-A71(Nagoya) (GenBank: AB482183), were used to evaluate the sensitivity of the DD methods. PV-positive stool samples from AFP cases in Laos and Cambodia, which were received between April 2021 and July 2022, and identified in the National Institute of Infectious Diseases as a part of the GPLN (a total of 10 samples), were used in this study. Six samples were positive for Sabin 3 and four samples were positive for both Sabin 1 and Sabin 3. All the clinical samples used in this study were appropriately anonymized; therefore, they were exempt from the regulation under the Committee for Ethical Regulation of the National Institute of Infectious Diseases after the review.

**RNA extraction.** A Zymo Quick-viral RNA kit (Zymo Research, Irvine, CA, USA) was used for RNA extraction according to the manufacturer’s instructions. Stool extracts were prepared from the PV-positive stool samples according to the WHO standard method (i.e., 2 g of stool sample added into 10 mL of PBS and 1 mL of chloroform). For the extraction of RNA from stool samples, 400 µL of the stool extracts mixed with 1200 µL of the Viral RNA buffer in the kit (a total of 1600 µL) was applied to the column and finally eluted in 20 µL of nuclease-free water [14]. For the extraction of RNA from virus solution, 200 µL of virus solution mixed with 400 µL of the Viral RNA buffer in the kit (a total of 600 µL) was applied to the column and finally eluted in 15 µL of nuclease-free water; then, 35 µL of nuclease-free water was added to give a total 50 µL of RNA solution.

**DD targeting the VP1 region.** The primers used in this study are shown in Table 1. The entire-capsid-coding region (approximately 3.9 kbp) was amplified by RT-PCR with 5′NTR-529-552+ (5′NTR) and Cre-4485-4460- (Cre) primers [5] (EC RT-PCR) using two commercial kits. The RT-PCR with a SuperScript III One-Step RT-PCR System with Platinum Taq DNA Polymerase (named kit A hereafter, Invitrogen, Carlsbad, CA, USA) was performed according to the procedure developed by Shaw et al. [6,15]. In this procedure, the 5′NTR primer was added after the RT reaction. The RT-PCR with a PrimeScript II High Fidelity One-Step RT-PCR Kit (named kit B hereafter, Takara Bio, Shiga, Japan) was optimized in the present study. The 5 µL of viral RNA solution was used in a total 25 µL reaction mixture. The RT-PCR conditions consist of an RT step at 45 °C for 15 min followed by inactivation at 94 °C for 2 min; 52 cycles of thermal cycling at 98 °C for 10 s, 55 °C for 15 s and 68 °C for 50 s; and a hold step at 10 °C. The 5′NTR and Cre primers were added together to the reaction mixture for the RT-PCR reaction. These RT-PCR products were diluted 100-fold with Milli-Q water; then, 1 µL of each diluted product was subjected to PV ITD rRT-PCR [16].

To obtain the cDNAs of VP1 region, the RT-PCR products were then subjected to a second PCR (nested PCR) with barcoded Y7/Q8 primers for nanopore sequencing or with type-specific primers (246S-S1/252A-S1 and 248S-S3/254A-S3 primers for the detection of Sabin 1 and Sabin 3, respectively) for Sanger sequencing. Non-barcoded Y7/Q8 primers were also used to evaluate the effect of the attached barcodes on the sensitivity of the detection. For the PCR reaction with barcoded primers, a DreamTaq DNA polymerase kit (Thermo Fisher Scientific, Waltham, MA, USA) was used according to the previous report [6,15]. With non-barcoded primers, an EmeraldAmp PCR Master Mix kit (Takara Bio, Shiga, Japan) was used. Both PCR reactions were performed with 2 µL of the first PCR products in a 25 µL of reaction mixture. The PCR conditions consist of a denaturation step at 95 °C for 2 min; 35 cycles of 95 °C for 30 s, 55 °C for 30 s and 72 °C for 1 min; and a hold step at 72 °C for 10 min [6,15]. For the PCR with the type-specific primers, a One-Step RT-PCR Kit (Qiagen, Venlo, The Netherlands) was employed, which is usually used for RT-PCR with viral RNA according to a manual by the WHO [3]. The RT-PCR conditions consist of an RT step at 50 °C for 30 min followed by inactivation at 95 °C for 15 min; 35 cycles of thermal cycling at 94 °C for 30 s, 45 °C for 30 s and 72 °C for 1 min; and an additional elongation step at 72 °C for 5 min. Raw images of gel electrophoresis are provided in Appendix A. The PCR products were analyzed by Sanger or nanopore sequencing.

**DD targeting the VP4–VP2 region.** The primers used in this study are shown in Table 1. The cDNAs of the VP4–VP2 regions of PV and non-polio enterovirus were amplified by RT-PCR [8,18,19] according to a protocol in the laboratory manual for pathogen detection of hand, foot and mouth disease (HFMD) in Japan [20]. The RT reaction was performed using a PrimeScript RT reagent Kit (Perfect Real Time) (Takara Bio, Shiga, Japan) with random hexamers (a total 10 µL reaction mixture). A semi-nested RT-PCR targeting the VP4–VP2 region was conducted with an EmeraldAmp PCR Master Mix (Takara Bio, Shiga, Japan). The 3 µL of RT products was used as the templates in the first PCR and 1 µL of the first PCR products was used as the templates in the second PCR in the reaction mixtures (a total 50 µL reaction mixture). EVP2 [18] and OL68-1 [19] primers were used for the first PCR, and EVP4 [8] and OL68-1 primers for the second PCR. The conditions of the first PCR consist of a denaturation step at 95 °C for 5 min; followed by 40 cycles of 95 °C for 30 s, 55 °C for 30 s and 72 °C for 45 s; and an additional elongation step at 72 °C for 5 min. The conditions of the second PCR consist of a denaturation step at 95 °C for 5 min; followed by 35 cycles of 95 °C for 30 s, 55 °C for 30 s and 72 °C for 45 s; and an additional elongation step at 72 °C for 5 min. Raw images of gel electrophoresis are provided in Appendix A. The PCR products were analyzed by Sanger sequencing.

**Sanger sequencing.** The PCR products were processed with an ExoSAP-IT Express (Thermo Fisher Scientific, Waltham, MA, USA) by mixing 5 µL of the products with 2 µL of ExoSAP and incubating the mixture at 37 °C for 4 min, 80 °C for 1 min and then at 4 °C. The products were subjected to sequencing reaction with a BigDye Terminator v3.1 cycle sequencing ready reaction kit (Applied Biosystems, Foster City, CA, USA) and then analyzed by a SeqStudio Genetic Analyzer (Applied Biosystems, Foster City, CA, USA). The 246S-S1 or 248S-S3 primers were used for the sequencing of the VP1 region. The EVP4/OL68-1 primers were used for the sequencing of the VP4–VP2 region. Sequence analysis was performed using GENETYX software ver. 16 (GENETYX Corp., Tokyo, Japan). For the analysis of the VP4–VP2 region, the amino acid sequences of the VP4–VP2 region were used for the type assignment using the National Center for Biotechnology Information’s (NCBI) Protein Basic Local Alignment Search Tool (blastp) search.

**Nanopore sequencing.** Nanopore sequencing was performed as previously reported [6,15]. The sequence library was prepared with an NEB Next Companion Module for Oxford Nanopore Technologies Ligation Sequencing and a SQK-LSK110 kit (Oxford Nanopore Technologies, Oxford, UK). Sequencing was performed on a MinION Mk1C sequencer using R9.4.1 flow cells (Oxford Nanopore Technologies, Oxford, UK). The data were analyzed by Poliovirus Investigation Resource Automating Nanopore Haplotype Analysis (PIRANHA) [21].

**Quantification of viral RNA.** The number of copies of the viral RNA was analyzed by real-time RT-PCR (rRT-PCR), as previously reported [22]. Viral RNA was assayed in a 20 µL reaction mixture containing 5 µL of viral RNA by using a One-Step SYBR PrimeScript PLUS RT-PCR Kit (Takara Bio, Shiga, Japan) with primers (EQ-1 and EQ-2, final concentration 0.8 μM each). Quantitated viral RNA of PV1(Mahoney) was used to control the quantification of the number of copies. The conditions of the rRT-PCR consist of an RT step at 42 °C for 30 min and 40 cycles of thermal cycling at 95 °C for 3 s and 60 °C for 30 s. The fluorescence emission of SYBR Green I was monitored and analyzed using an Applied Biosystems 7500 Fast Real-Time PCR System (Applied Biosystems, Foster City, CA, USA).

**ITD.** PV ITD rRT-PCR was performed using a Quant Studio 5 Real-Time PCR (Applied Biosystems, Foster City, CA, USA) with a qScript XLT One-Step RT-qPCR ToughMix kit (Quanta) and primer/probe sets provided in the PV Diagnostic rRT-PCR kit (ver. 5.0, US CDC) [23]. One µL of diluted EC RT-PCR products (diluted 100-fold with water) was added to 19 µL of the RT-PCR reaction mix and then subjected to rRT-PCR. PV ITD rRT-PCR conditions consist of an RT step at 50 °C for 30 min followed by inactivation at 95 °C for 1 min; then, 40 cycles of thermal cycling at 95 °C for 15 s, 50 °C for 45 s and 72 °C for 5 s (25% ramp rate from 50 °C to 72 °C).

## 3. Results

### 3.1. DD Targeting the VP1 Region

A flowchart of the DD methods evaluated in this study is shown in Figure 1. For the DD targeting the VP1 region, the cDNA of the entire-capsid-coding region (approximately 3.9 kbp) was amplified by EC RT-PCR (the first PCR) [5]. Then, the cDNA of the VP1 region was amplified by using the RT-PCR products as templates with barcoded Y7/Q8 primers (for nanopore sequencing) or type-specific VP1 primers (for Sanger sequencing) (pan-PV-VP1 PCR, the second PCR).

First, we evaluated the limit of detection (LOD) of the DD methods with quantitated PV RNAs extracted from the virus stock solution (Figure 2A). In the evaluation, the performance of two commercially available kits (kits A and B) and the effect of barcodes on the primers were also analyzed. The LOD of EC RT-PCR was 820 copies (about 10 CCID_50_) or 82 copies (about 1 CCID_50_) with kits A and B, respectively. The LODs after the pan-PV-VP1 PCR (the second PCR) were 82 copies (about 1 CCID_50_) for both kits (used in the EC RT-PCR, the first PCR), irrespective of the presence of barcodes on the primers. The RT-PCR reaction by kits A and B took approximately 5 and 2.5 h, respectively.

Next, we attempted to detect PV from the PV-positive stool samples by the DD methods (Figure 2B). From these stool samples, PVs were isolated at day 4 post-inoculation or during the second passage (Table 2), suggesting that the viral titers in the samples were relatively low. The number of copies of viral RNA in 200 µL of the stool suspension was estimated from the extracted RNA solution, which ranged from 18 to 87,000. The detection rates by the EC RT-PCR were 40% (4/10) and 80% (8/10) with kits A and B, respectively. The detection rates after the pan-PV-VP1 PCR were 60% (6/10) and 80% (8/10) with the RT-PCR products obtained using kits A and B, respectively. ITD was performed with the RT-PCR products obtained using kits A or B (Table 2). Mixtures of Sabin 1 and Sabin 3 were detected for the products obtained using kit A but not for those obtained using kit B, suggesting that the efficiency of the RT-PCR could negatively affect correct identification of virus mixtures. The observed discrepancy between the results of VI and ITD suggests a high sensitivity of ITD that could even detect non-viable viral genomes [5,24].

In the pan-PV-VP1 PCR, barcoding of the primers had no effect on the detection rates. The detection rates with type-specific primers were similar to those with pan-PV primers; however, only one sample (sample 8) was positive for Sabin 1 and Sabin 3 (with the kit A) among the four stool samples with the mixtures. The type-specific-VP1 PCR showed a correlation with the results of ITD (Figure 2 and Table 2); however, the mixtures of PV were correctly detected only from one sample with kit A.

We performed sequence analysis for the products of the pan-PV-VP1 PCR or of type-specific-VP1 PCRs (Table 2). By nanopore sequencing, only one PV strain was detected for each sample: Sabin 1 or Sabin 3. The signals of the sequencing were only obtained from the PCR-positive samples; no signals were obtained from the PCR-negative samples. By Sanger sequencing, three samples were positive for both Sabin 1 and Sabin 3. These results suggested that the sequencability of the DD methods was primarily determined by the efficiency of the PCRs (i.e., the EC RT-PCR, pan-PV-VP1 PCR and type-specific-VP1 PCR) and that PV strains in the mixture could be missed by the DD methods.

### 3.2. DD Targeting the VP4–VP2 Region

For the DD targeting the VP4–VP2 region, a semi-nested RT-PCR followed by Sanger sequencing was performed [8,18,19,20]. The LOD of the VP4–VP2 first PCR for PV was 250 copies (about 3 CCID_50_) (Figure 3A) and that after the VP4–VP2 second PCR was 25 copies (about 0.3 CCID_50_). Meanwhile, the LOD after the VP4–VP2 second PCR for EV-A71(Nagoya) and EV-D68(US/KY/14-18953) were 6600 copies (33 CCID_50_) and 42 copies (2.0 × 10^−3^ CCID_50_), respectively. The RT-PCR reaction took approximately 3 h.

We attempted to detect PV from the stool samples with the DD targeting the VP4–VP2 region. The PCR was positive for all PV-positive stool samples (Figure 3B). PV was identified by sequencing analysis from all the samples; however, only one PV strain could be identified for each sample with PV mixtures (Table 3). This suggested that the sensitivities of the DD targeting the VP4–VP2 region might be close to those of VI; however, PV strains in the mixtures could not be identified by this DD method.

## 4. Discussion

In the present study, we evaluated protocols of previously reported DD methods in terms of their sensitivity and sequencability. For the DD targeting the VP1 region, two commercially available kits were examined. We found that the sensitivity of the PrimeScript II High Fidelity One-Step RT-PCR Kit (named kit B in this study) was slightly higher than that of the SuperScript III One-Step RT-PCR System with Platinum Taq DNA Polymerase (named kit A in this study) (detection rates of 80% vs. 40% in the EC RT-PCR). The optimized protocol for kit B was simpler and the reaction was faster than those of kit A; with the addition of all the primers at the start of reaction, the RT-PCR reaction finished within 2.5 h; however, the efficient amplification could negatively affect the detection rate of minor populations of PV strains in the mixtures (Table 2, see below discussion). Selection of commercially available kits is apparently a critical factor for the performance of the DD, including the extraction step of viral RNA [25,26].

The detection rates of PV by the DD methods targeting the VP1 region have yet to reach that of VI (about 60 to 80% that of VI, Figure 2B). This may make an apparent discrepancy with a recent report on DD that used the same methodology in large-scale surveillance and showed an almost comparable detection rate to VI [27]. It should be noted that there is no chance that DD and VI provide 100% concordant results since DD can detect even non-infectious genomes, besides the intrinsic difference of the sensitivity. In contrast to the method targeting the VP1 region, the detection rate of the DD method targeting the VP4–VP2 region was 100% (Figure 3B). The LODs of the DD methods targeting the VP1 region were 82 copies for PV genomes and that of the DD method targeting the VP4–VP2 region was 25 copies. Partial fragmentation of the extracted viral RNA and/or an intrinsic limitation of the long PCR might have acted as the causes of this difference. The LOD of 25 copies of the PV genome might serve as an index to evaluate the potency of the DD methods. The slightly low detection rates of the DD methods targeting the VP1 region may be overcome by simply increasing the amounts of the RNA available for the test; other factors than the amount of RNA, including inhibitors and competitors of RT-PCR reaction in the extracted RNA and the quality of RNA, may pose another challenge to improve the detection rates.

We could not correctly identify PVs in the stool samples that contain mixtures of Sabin 1 and Sabin 3 by the DD methods, which may possibly be linked to a vaccination event with a bivalent OPV that consists of Sabin 1 and Sabin 3 rather than to a long-term circulation. In the pan-PV-VP1 PCR, while the barcoding of the primers had no effect on the detection rates, which is essential for high-throughput analysis, the minor populations of PV strains could be missed after the RT-PCR (Table 2), possibly due to dominant amplification of the cDNA of the major population. The ITD with the RT-PCR products was quite sensitive and detected minor populations of PV strains, which were negative by VI. These PV strains could be either infectious or noninfectious, as previously observed [5,24]. The PCR with type-specific primers improved the detection rates, consistent with a recent report on VDPV detection from environmental samples [28]. A mixture of PV strains was detected only by type-specific-VP1 PCR followed by Sanger sequencing but not by pan-PV-VP1 PCR followed by nanopore sequencing. Therefore, for the DD methods targeting the VP1 region, ITD with the EC RT-PCR products seemed essential to maximize the efficiency of PV identification with type-specific primers. For the DD targeting the VP4–VP2 region, analyses of both the mixtures and the ITD system remain to be established. Collectively, identification of minor populations of PV strains in virus mixtures remained a substantial challenge in the DD methods in the surveillance because PV/EV species C mixtures could be frequently observed in AFP cases in Africa [29].

Limitations of this study include the number of PV-positive stool samples available for the test (only 10 samples); however, some stool samples seemed to contain an almost minimal amount of PV detectable by VI (only detected after second passage, Table 2). Therefore, the results obtained from these stool samples would not unreasonably overestimate of the performance of the DD methods with general PV-positive stool samples.

In summary, we evaluated the sensitivity and sequencability of reported DD methods targeting the VP1 or VP4–VP2 regions. The DD targeting the VP4–VP2 region would be useful for the detection of PV, especially in emergency response, which needs quick results to determine PV-positive or -negative patients (e.g., as a diagnostic mean for accidentally PV-exposed persons or for early outbreak response to estimate the scale or area of the outbreak) due to the high sensitivity. However, the DD targeting the VP4–VP2 region may not provide sufficient information to track the origin of the PV isolates, in part due to the short length. The DD targeting the VP1 region remained a challenge in terms of sensitivity and posed tremendously more challenges in the context of virus mixtures to serve as an alternative to VI in the surveillance.

## Figures and Tables

**Figure 1 viruses-15-02113-f001:**
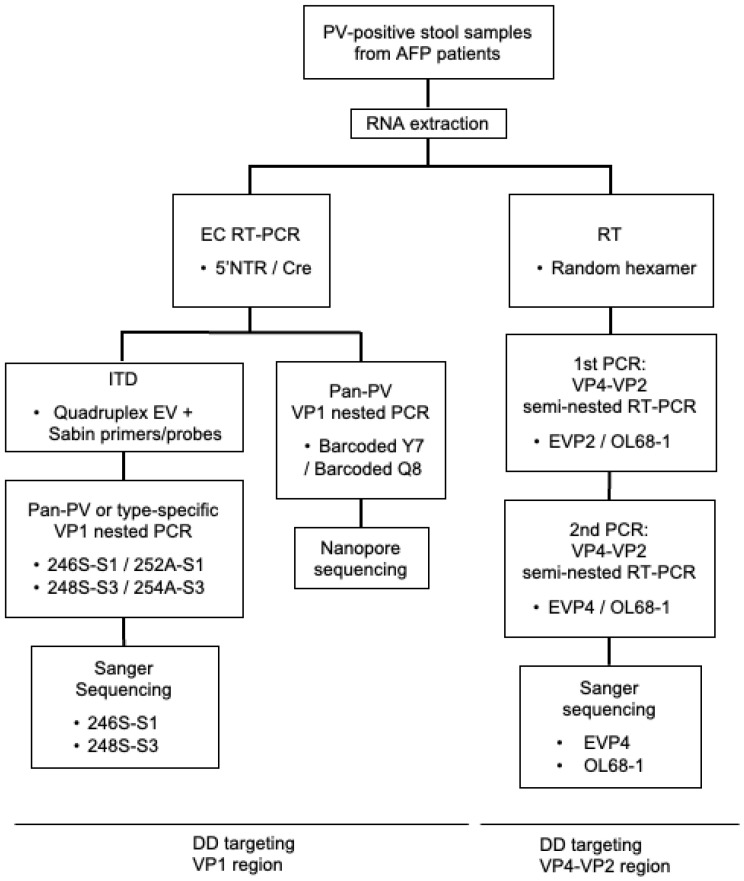
**Flowchart of direct detection (DD) methods for PV.** Viral RNA was extracted from stool suspensions that were positive for PV by VI. Extracted RNAs were subjected to DD methods targeting the VP1 region either by Sanger sequencing or nanopore sequencing or to a DD method targeting the VP4–VP2 region by Sanger sequencing.

**Figure 2 viruses-15-02113-f002:**
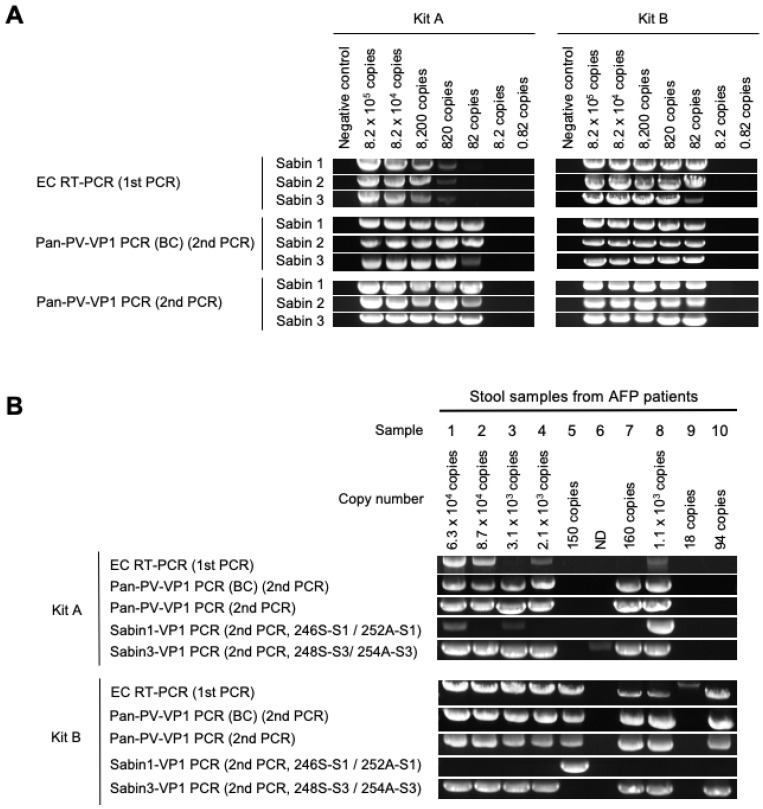
**Sensitivity of DD methods targeting the VP1 region.** (**A**) Sensitivity of the DD methods for PV genomes. (**B**) PV detection from PV-positive stool samples. EC RT-PCR was performed using kits A or B with indicated number of copies of viral genomic RNAs of PV Sabin strains (Sabin 1, Sabin 2 and Sabin 3 strains) or viral RNAs extracted from stool suspensions. Pan-PV-VP1 PCR was performed with barcoded (BC) or non-barcoded Y7/Q8 primers. Sabin 1/Sabin 3-VP1 PCR was performed with type-specific primers. Estimated numbers of copies of the viral genome included in 200 µL of each stool suspension are shown. ND, not detected.

**Figure 3 viruses-15-02113-f003:**
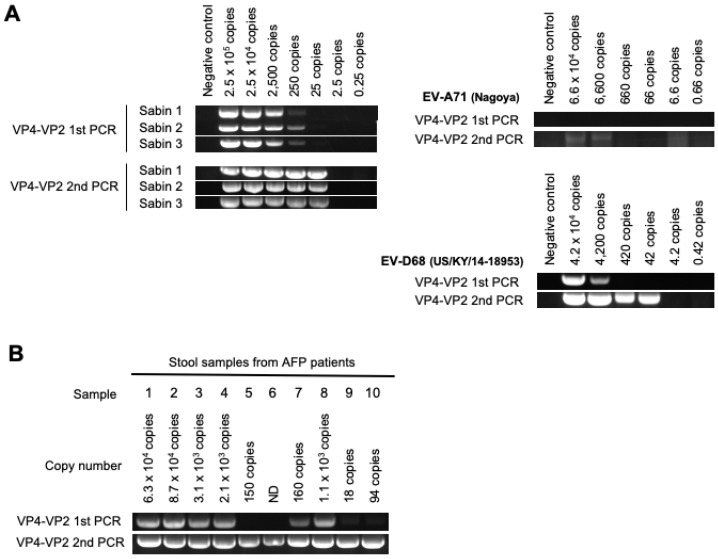
**Sensitivity of a DD method targeting the VP4–VP2 region.** (**A**) Sensitivity of the DD method for PV, EV-A71 and EV-D68 genomes. Semi-nested PCR (VP4–VP2 first PCR and second PCR) was performed with the indicated number of copies of viral genomes of PV Sabin strains, EV-A71 (Nagoya) and EV-D68 (US/KY/14-18953). (**B**) PV detection from PV-positive stool samples. Estimated numbers of copies of the viral genome included in 200 µL of each stool suspension are shown. ND, not detected.

**Table 1 viruses-15-02113-t001:** List of primers.

Detection Method	Primer	Sequence (5′–3′)	Gene	Position *	Reference
EC RT-PCR	5′NTR-529-552+ (5′NTR)	TGGCGGAACCGACTACTTTGGGTG	5′NTR	529–552	[5]
	Cre-4485-4460- (Cre)	TCAATACGGTGTTTGCTCTTGAACTG	2C	4460–4485	
Pan-PV-VP1 PCR	Y7	GGGTTTGTGTCAGCCTGTAATGA	VP3	2399–2421	[17]
	Q8	AAGAGGTCTCTRTTCCACAT	2A	3485–3504	
Type-specific VP1 PCR	246S-S1	CGAGATACCACACATATAGA	VP3	2441–2460	[17]
	248S-S3	CGAGACACCACTCACATTTC	VP3	2438–2457	
	252A-S1	ATATGTGGTCAGATCCTTGGTG	VP1	3364–3385	
	254A-S3	ATATGTGGTTAATCCTTTCTCA	VP1	3355–3376	
VP4–VP2 semi-nested RT-PCR	EVP2	CCTCCGGCCCCTGAATGCGGCTAAT	5′NTR	444–468	[18]
	EVP4	CTACTTTGGGTGTCCGTGTT	5′NTR	541–560	[8]
	OL68-1	GGTAAYTTCCACCACCANCC	VP2	1178–1197	[19]

* Position of PV1(Mahoney) (Accession number: V01149) except for 248S-S3 and 254A-S3. The positions for 248S-S3 and 254A-S3 are of PV3(Sabin 3) (Accession number: AY184221).

**Table 2 viruses-15-02113-t002:** PV VP1 sequencing results by Sanger and nanopore sequencing.

			Samples											
			1	2	3	4	5	6	7	8	9	10		
		Number of copies of viral genome in 200 μL of stool suspension	6.3 × 10^4^	8.7 × 10^4^	3.1 × 10^3^	2.1 × 10^3^	150	ND	160	1.1 × 10^3^	18	94		
RT-PCR kit	Method												Detection rate	Identification rate *
	VI		Sabin 3 (L20B, 2nd passage) **	Sabin 3 (L20B, 2nd passage)	Sabin 3 (L20B, day 4)	Sabin 3 (L20B, day 4)	Sabin 1 + Sabin 3 (L20B, day 4)	Sabin 1 + Sabin 3 (L20B, day 4)	Sabin 1 + Sabin 3 (L20B, day 4)	Sabin 1 + Sabin 3 (L20B, day 4)	Sabin 3 (RD, 2nd passage)	Sabin 3 (RD, 2nd passage)		
Kit A	EC RT-PCR + ITD	Ct value for Sabin 1	29.0	29.3	28.3	29.1	29.4	28.3	28.8	20.6	29.4	29.6	100% (10/10)	
		Ct value for Sabin 3	10.6	11.3	13.5	13.3	-	30.3	17.5	11.9	32.9	-	80% (8/10)	
	Nanopore sequencing		Sabin 3	Sabin 3	Sabin 3	Sabin 3	-	-	Sabin 3	Sabin 3	-	-	60% (6/10)	40% (4/10)
	Sanger sequencing		Sabin 1 + Sabin 3	Sabin 3	Sabin 1 + Sabin 3	Sabin 3	-	Sabin 3	Sabin 3	Sabin 1 + Sabin 3	-	-	70% (7/10)	30% (3/10)
Kit B	EC RT-PCR + ITD	Ct value for Sabin 1	-	-	-	-	10.0	-	-	-	-	-	10% (1/10)	
		Ct value for Sabin 3	8.6	6.0	6.3	6.4	-	-	9.8	10.1	-	10.5	70% (7/10)	
	Nanopore sequencing		Sabin 3	Sabin 3	Sabin 3	Sabin 3	Sabin 1	-	Sabin 3	Sabin 3	-	Sabin 3	80% (8/10)	50% (5/10)
	Sanger sequencing		Sabin 3	Sabin 3	Sabin 3	Sabin 3	Sabin 1	-	Sabin 3	Sabin 3	-	Sabin 3	80% (8/10)	50% (5/10)

* Samples identified as “Sabin 1” or “Sabin 3” by DD + sequencing, although identified as “Sabin 1 + 3” by VI + ITD, were regarded as not identified. ** Date of PV isolation in cell culture (L20B or RD cells) in 1st passage or 2nd passage. ND, not detected.

**Table 3 viruses-15-02113-t003:** PV VP4–VP2 sequencing results by Sanger sequencing.

		Sample											
		1	2	3	4	5	6	7	8	9	10		
	Number of copies of viral genome in 200 μL of stool suspension	6.3 × 10^4^	8.7 × 10^4^	3.1 × 10^3^	2.1 × 10^3^	150	ND	160	1.1 × 10^3^	18	94		
Method												Detection rate	Identification rate *
VI		Sabin 3	Sabin 3	Sabin 3	Sabin 3	Sabin 1 + Sabin 3	Sabin 1 + Sabin 3	Sabin 1 + Sabin 3	Sabin 1 + Sabin 3	Sabin 3	Sabin 3		
Sanger sequencing		Sabin 3	Sabin 3	Sabin 3	Sabin 3	Sabin 1	Sabin 1	Sabin 3	Sabin 3	Sabin 3	Sabin 3	100% (10/10)	60% (6/10)

* Samples identified as “Sabin 1” or “Sabin 3” by DD + sequencing, although identified as “Sabin 1 + 3” by VI + ITD, were regarded as not identified. ND, not detected.

## Data Availability

Raw data sets not included in this paper are available from the corresponding authors upon request.

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
