# Peer review of "Evaluation of Direct Detection Protocols for Poliovirus from Stool Samples of Acute Flaccid Paralysis Patients"

_viruses, 2023, doi:10.3390/v15102113_

Round 1

Reviewer 1 Report

Until now the gold standard method for detecting poliovirus has been viral isolation. However, this method is time consuming and requires a lot of resources and know-how that are difficult to maintain in the global polio surveillance network which includes more than 100 laboratories. For decades, molecular methods for direct detection of poliovirus in stool samples from people with paralysis have been developed. However, the sensitivity and reliability of the proposed methods have so far proven to be insufficient.

This article describes two RT-PCR methods which considerably advance research in this domain, one of them showing detection rates very close to those of the standard method, except for the detection of mixtures of viruses of different types. Despite this limitation, this work as well as that recently published by Shaw et al. (ref 26), constitute significant advances essential to make the final necessary improvements which will allow the development of a molecular detection test for poliovirus at least equivalent to the standard WHO detection test.

Minor comment:

Line 86: How were the stool extracts prepared? Is the method identical to that proposed in the WHO standard method? What is the ratio of feces material weight to buffer volume?

Author Response

Thank you very much for your generous and constructive comments to improve the manuscript. We modified the text according to the reviewers’ suggestions, and also corrected grammatical errors in the text. Modified parts were shown with track markup.

Reviewer 1:

Until now the gold standard method for detecting poliovirus has been viral isolation. However, this method is time consuming and requires a lot of resources and know-how that are difficult to maintain in the global polio surveillance network which includes more than 100 laboratories. For decades, molecular methods for direct detection of poliovirus in stool samples from people with paralysis have been developed. However, the sensitivity and reliability of the proposed methods have so far proven to be insufficient.

This article describes two RT-PCR methods which considerably advance research in this domain, one of them showing detection rates very close to those of the standard method, except for the detection of mixtures of viruses of different types. Despite this limitation, this work as well as that recently published by Shaw et al. (ref 26), constitute significant advances essential to make the final necessary improvements which will allow the development of a molecular detection test for poliovirus at least equivalent to the standard WHO detection test.

Answer:

Thank you very much for your generous comment. We modified the text according to the suggestion.

Minor comment:

Line 86: How were the stool extracts prepared? Is the method identical to that proposed in the WHO standard method? What is the ratio of feces material weight to buffer volume?

Answer:

Thank you very much for your kind suggestion. We clarified the preparation of the stool extracts in the text.

Reviewer 2 Report

In this article, the authors evaluate the performances of different molecular strategies that aim to detect poliovirus genomes in stools and to allow the sequencing of the genomic region (namely the VP1-encoding 1D region) used to characterize field polioviruses. This topic is of great interest for the polio eradication program. Please find my comments and suggestions below.

1-    Line 86. The authors indicate that 400 µL of stool extracts were applied to the column. By “stool extract”, do they mean “stool suspension”, i.e., stool mixed with PBS? Could the authors provide more details about the procedures followed to prepare this “stool extract”?

2-    Line 88. “200 µL of virus solution was applied to the column”. I imagine it means that 200 µL of virus solution was mixed with the same volume of lysis buffer. It would be better to clarify to avoid misunderstanding.

3-    Line 89. RNAs are extracted with 15 µL of water and then 35 µL of water was added to the extracts. Why not to elute with 50 µL? Did the authors compare both elution volumes?

4-    Lines 229-230. This sentence is not clear to me. Does it mean (as suggested by Figure 1) that the ITD was performed by using as sample the RT-PCR products (i.e., DNA) generated through the EC RT-PCR? If yes, how to use RT-PCR kit and program (Line 189-195)?

5-    In tables 2 & 3, the column “Detection rate” is not clear to me. How can a rate be higher than 100%? Besides, when a sample contains 2 viruses, this rate should not be 100% if only one virus is found in each sample (since one is missed).

6-    I slightly disagree with the lines 320-322. Increasing the amounts of RNA is not that simple because a- there is a maximum volume that can be added in the RT-PCR reaction and b- adding RA also mean adding inhibitors and competitors that could impair the amplification. 

7-    I feel that the discussion is rather a summary of the results than a real discussion. I think the authors should emphasize the main results of their study: 1- There is no chance that DD and VI are 100% concordant since DD can detect non-infectious genomes; 2- Barcoding does not reduce the sensitivity of the RT-PCR (which is an important result for laboratories that deal with numerous samples); 3- Detecting PV genomes by DD is already a challenge but getting sequences is tremendously more challenging, especially in case of mixtures. The authors could place these conclusions in the context of the polio program and identify contexts where DD could be useful and others where its limitations would make it ineffective; in particular, what place do the authors think VP4-VP2 screening could occupy in the GPLN algorithm in the future?

Author Response

Thank you very much for your generous and constructive comments to improve the manuscript. We modified the text according to the reviewers’ suggestions, and also corrected grammatical errors in the text. Modified parts were shown with track markup.

Reviewer 2:

In this article, the authors evaluate the performances of different molecular strategies that aim to detect poliovirus genomes in stools and to allow the sequencing of the genomic region (namely the VP1-encoding 1D region) used to characterize field polioviruses. This topic is of great interest for the polio eradication program. Please find my comments and suggestions below.

Answer:

Thank you very much for your generous comment. We modified the text according to the suggestions.

  • Line 86. The authors indicate that 400 µL of stool extracts were applied to the column. By “stool extract”, do they mean “stool suspension”, i.e., stool mixed with PBS? Could the authors provide more details about the procedures followed to prepare this “stool extract”?

Answer:

Thank you very much for your kind suggestion. We clarified the preparation of the stool extracts in the text below. We may prefer ‘stool extract’ in this manuscript because of the chloroform treatment of the suspension.

: Stool extracts were prepared from the PV-positive stool samples according to the WHO standard method (i.e., 2 g of stool sample added into 10 mL of PBS and 1 mL of chloroform).

  • Line 88. “200 µL of virus solution was applied to the column”. I imagine it means that 200 µL of virus solution was mixed with the same volume of lysis buffer. It would be better to clarify to avoid misunderstanding.

Answer:

Thank you very much for your kind suggestion. We clarified the preparation of the RNA extraction in the text.

  • Line 89. RNAs are extracted with 15 µL of water and then 35 µL of water was added to the extracts. Why not to elute with 50 µL? Did the authors compare both elution volumes?

Answer: Thank you very much for your kind suggestion. For the RNA extraction from the virus solution, we extracted the RNA according to the manufacturer’s instruction (elution with 15 uL of water). To facilitate the quantification of viral RNA in the eluates in this study, we added 35 uL of water to the eluates.

  • Lines 229-230. This sentence is not clear to me. Does it mean (as suggested by Figure 1) that the ITD was performed by using as sample the RT-PCR products (i.e., DNA) generated through the EC RT-PCR? If yes, how to use RT-PCR kit and program (Line 189-195)?

Answer: Thank you very much for your kind suggestion. As suggested in this comment, we used DNA (EC RT-PCR products diluted by 100-fold with water) for the ITD. Therefore, the RT step might not be essential for the measurement of these samples, but we included this step in the measurement so as not to disturb the whole process of this kit. We clarified this point in the text.

  • In tables 2 & 3, the column “Detection rate” is not clear to me. How can a rate be higher than 100%? Besides, when a sample contains 2 viruses, this rate should not be 100% if only one virus is found in each sample (since one is missed).

Answer: Thank you very much for your kind suggestion. The original definition of this detection rate was relative detection rate to the results of the virus isolation. But as suggested in this comment, we agree that the definition might not be intuitively clear and could be inconsistent between the tests. Therefore, we corrected the values in Table 2 (the total number of sample (thus 10) was taken as 100%) to clarify the definition of the detection rate.

6-    I slightly disagree with the lines 320-322. Increasing the amounts of RNA is not that simple because a- there is a maximum volume that can be added in the RT-PCR reaction and b- adding RNA also mean adding inhibitors and competitors that could impair the amplification. 

 Answer: Thank you very much for your kind suggestion. We agree with the reviewer’s concern about this challenge and our optimism. We added a sentence in the text as below:

; other factors than the amount of RNA, including inhibitors and competitors of RT-PCR reaction in the extracted RNA and the quality of RNA might pose another challenge to improving the detection rates.

7-    I feel that the discussion is rather a summary of the results than a real discussion. I think the authors should emphasize the main results of their study: 1- There is no chance that DD and VI are 100% concordant since DD can detect non-infectious genomes; 2- Barcoding does not reduce the sensitivity of the RT-PCR (which is an important result for laboratories that deal with numerous samples); 3- Detecting PV genomes by DD is already a challenge but getting sequences is tremendously more challenging, especially in case of mixtures. The authors could place these conclusions in the context of the polio program and identify contexts where DD could be useful and others where its limitations would make it ineffective; in particular, what place do the authors think VP4-VP2 screening could occupy in the GPLN algorithm in the future?

Answer: Thank you very much for your kind suggestion. We agree with the suggestions. We clarified these points in the Discussion section including the possible utility of the methods.

Reviewer 3 Report

In this article the authors  presented the results of the  evaluation of the sensitivity and sequencability of reported  DD methods targeting the VP1 or VP4-VP2 region. In the emergency  situation the DD targeting the VP4-VP2 region would be useful for the detection of PV due to the high sensitivity. 

For the DD targeting the VP1 region, improvement of the sensitivity and the detection of minor population of PV strain in the virus mixtures remained  an alternative to the VI. The results are very important in the context of the Global Polio Erdication Strategy and GAP IV. 

Author Response

Thank you very much for your generous comment.